# Representative Exposure–Annoyance Relationships Due to Transportation Noises in Japan

**DOI:** 10.3390/ijerph182010935

**Published:** 2021-10-18

**Authors:** Shigenori Yokoshima, Makoto Morinaga, Sohei Tsujimura, Koji Shimoyama, Takashi Morihara

**Affiliations:** 1Kanagawa Environmental Research Center, Hiratsuka 254-0014, Japan; 2Department of Architecture, Faculty of Engineering, Kanagawa University, Yokohama 221-8686, Japan; m-morinaga@kanagawa-u.ac.jp; 3Graduate School of Science and Engineering, Ibaraki University, Hitachi 316-8511, Japan; sohei.tsujimura.fifty@vc.ibaraki.ac.jp; 4Aviation Environment Research Center, Organization of Airport Facilitation, Tokyo 105-0011, Japan; k-shimoyama@aeif.or.jp; 5National Institute of Technology, Ishikawa College, Tsubata 929-0392, Japan; morihara@ishikawa-nct.ac.jp

**Keywords:** transportation noise, exposure–response relationship, annoyance, secondary analysis

## Abstract

This paper focuses on clarifying the relationship between noise exposure and the prevalence of highly annoyed people due to transportation noise in Japan. The authors accumulated 34 datasets, which were provided by Socio-Acoustic Survey Data Archive and derived from the other surveys conducted in Japan. All the datasets include the following micro-data: demographic factors, exposure, and annoyance data associated with specific noise sources. We performed secondary analyses using micro-data and established the relationships between noise exposure (*L*_den_) and the percentage of highly annoyed people (%HA) for the following noise source: road traffic, conventional railway, Shinkansen railway, civil aircraft, and military aircraft noises. Among the five transportation noises, %HA for the military aircraft noise is the highest, followed by civil aircraft noise and Shinkansen railway noise. The %HA for conventional railway noise was higher than that for road traffic noise. To validate the representativeness of the exposure–response curves, we have discussed factors affecting the difference in annoyance. In addition, comparing the Japanese relationship with that shown in the “Environmental Noise Guidelines for the European Region,” we revealed that Japanese annoyance is higher than the WHO-reported annoyance.

## 1. Introduction

The “Environmental Noise Guidelines for the European Region” (Guidelines) [1] published in 2018 shows the guideline exposure levels (*L*_den_) for each noise source. In the guidelines, a meta-analysis of surveys conducted since 2000 was performed. Based on the exposure–annoyance relationship obtained by the meta-analysis, the Guideline Development Group set the guideline exposure level (*L*_den_) for daily average noise exposure. The guideline exposure levels for road traffic, railway and aircraft noises are determined to be 53 dB, 54 dB, and 45 dB, respectively.

Moving on to standards for environmental noises in Japan, the Environmental Quality Standards (EQS) are defined as the standards whose maintenance is desirable for the preservation of the living environment and are conducive to the protection of human health. At present, the following three EQSs are legislated: noise (general residential areas and areas facing roads), aircraft noise, and Shinkansen super-express railway noise. As per the EQSs for Noise, the standard values of noise in areas facing trunk roads are 70 dB and 65 dB *L*_Aeq_ in the daytime (6:00–22:00) and nighttime (22:00–6:00), respectively. Converting the standard values of the EQS for Noise comes out to about 74 dB in *L*_den_. Here, *L*_den_ is calculated based on the following time category: daytime, 7:00–19:00; evening, 19:00–22:00; and nighttime, 22:00–7:00. In the case of the EQSs for aircraft noise, the standard values in areas used exclusively for residential purpose are 57 dB in *L*_den_. On the other hand, the EQSs for Shinkansen super-express railway noise are evaluated not in an energy-based index but in a maximum-based index. The standard value is 70 dB in areas used mainly for residential purposes. Considering the frequency of trains in operation at the Tokaido Shinkansen Line, with trains operating most frequently at the Shinkansen Lines, the standard value is estimated to be around 57 dB in *L*_den_. However, the EQSs for the conventional railway are not legislated. These EQS values are higher than the corresponding guideline exposure levels. Taking strong recommendations, Japanese government is currently discussing whether the guideline exposure levels can be applied to standard values for the EQSs for the noises or not.

Large numbers of social surveys on community response to environmental noises, such as road traffic, railway, and aircraft noises, have been carried out in Japan. These surveys obtained micro-data consisting of noise exposure data and reactions to noise by respondents. While exposure–response relationships determined from the results of such socio-acoustic surveys can form the basis of an effective noise policy, the micro-data has yet to be accumulated into a unified system. Thus, these scientific findings based on the surveyed data have not effectively contributed to reviewing or creating noise policy. The Institute of Noise Control Engineering/Japan (INCE/J) set up a technical subcommittee on Socio-Acoustic Survey Data Archive (SASDA) in 2009. The subcommittee members and the Ministry of the Environment in Japan (MOE) deposited their own surveyed data. After checking the data, the subcommittee has been managing the Socio-Acoustic Survey Data Archive since 2011. The procedures for operating the SASDA, such as deposit, access, publication, and maintenance, were reported [2].

Using the dataset stored in SASDA the technical committee performed a secondary analysis. It showed the percentage of highly annoyed people (%HA) as a function of *L*_dn_ for each transportation noise (road traffic, conventional railway, Shinkansen railway, and military aircraft and civil aircraft noises) in Japan [3,4]. It should be noted that no socio-acoustic survey associated with industrial noise was carried out. The number of collected micro-data used for the analysis exceeds 20,000. In this paper, adding the other datasets derived from recent studies conducted after 2007 and reconsidering the calculation method of 72%HA according to the Guidelines, we performed a secondary analysis using more than 30,000 micro-data.

The purpose of this study is to establish the representative relationships between *L*_den_ and estimated %HA for each transportation noise source. This is the first paper establishing exposure–annoyance relationship associated with every transportation noise in Japan. This paper has been divided into five parts, including this introductory section. The second section is concerned with the materials and methodology used for this study. The third section gives an overview of the exposure–annoyance relationship for each of 34 datasets. Moreover, we investigated the effects of demographic factors and housing type on %HA. Based on the above considerations, extracting micro-data with the annoyance measured on a five-point verbal scale, which is proposed by ICBEN [5] and adopted in the International Organization for Standardization/technical specifications (ISO/TS) 15666 [6], we established the representative relationships between *L*_den_ and estimated %HA for each transportation noise source in Japan. In addition, we compared the Japanese representative relationship with that shown in the Environmental Noise Guidelines for the European Region. The fourth section discusses the results. Finally, the conclusion gives a summary and critiques of the findings.

## 2. Materials and Methods

### 2.1. Datasets

The 34 datasets used for analyses in this paper are shown in Table 1. Abbreviations of RT, CR, HR, CA, MA, and CB denote respective noises of road traffic, conventional railway, high-speed railway (Shinkansen railway), civil aircraft, military aircraft, and combined noises. Here, military aircraft noise means noises generated by aircraft noise from the Japan Self-Defense Forces and/or U.S. Forces. Sample size shows the number of data that are valid for both noise exposure and annoyance rating. The sample size for each noise source is 6779, 10488, 7094, 1469, and 4847 for RT, CR, HR, CA, and MA, respectively.

The 34 datasets were derived from 31 socio-acoustic surveys conducted from 1994 to 2017 in Japan, and the survey sites were wide-ranged in Japan. Numbers 1 to 23 are datasets stored in SASDA, and subsequent datasets, numbers from 24 to 34, are not stored yet. It should be noted that the authors have given the numbers of datasets from 24 to 34 for convenience.

Twenty-eight of the thirty-four datasets are derived from surveys conducted in areas exposed to a single specific noise. On the other hand, JPN021RT2004 and JPN021CR2004 are derived from a survey conducted in areas exposed to combined road traffic and conventional railway noises. In addition, KMM103CR2011 and KMM103HR2011 are derived from a survey conducted in areas exposed to Shinkansen and conventional railway noises. Similarly, KMM108CR2016 and KMM108HR2016 are datasets with the same combined noise sources. Annoyance due to each noise was obtained from one respondent in the surveys.

In the 1990s, many surveys addressed community response to road traffic noise. However, the number of surveys has decreased since 2000. Between 2000 and 2006, MOE carried out socio-acoustic surveys for each of the transportation noises in many areas around the nation. The datasets of JPN011RT2000, JPN017CR2003, JPN018HR2003, JPN019CA2003, and JPN020MA2003 were derived from a series of surveys. In addition, surveys aimed at constructing the noise annoyance scale in Japanese were performed (JPN012CR2001 and JPN014CR2002).

In recent years, before and after the opening of new projected Shinkansen lines, the Kyushu and the Hokuriku Shinkansen lines, socio-acoustic surveys along the lines have been vigorously carried out. The datasets related to the former surveys are KMM102CR2009, KMM103CR2011, KMM103HR2011, KMM108CR2016, and KMM108HR2016 in Kumamoto prefecture. The datasets associated with the latter surveys are NGN105HR2013 and HKR107HR2016 in Nagano prefecture, and Ishikawa and Toyama prefectures, respectively.

However, few surveys were conducted in the areas mainly exposed to civil or military aircraft noise. Only three and two datasets used for analysis are available for civil and military aircraft noise, respectively. This might be due to the fact that only a few surveys were conducted, and micro-data was difficult to extract from them.

### 2.2. Descriptor of Annoyance and Number of Scale Points

Table 2 shows the number of scale points and descriptor of annoyance due to the specific noise source used in each dataset.

After 2000, eighteen datasets included annoyance measured with a verbal scale, using modifiers (“not at all”, “slightly”, “moderately”, “very”, and “extremely”) and descriptors (“bothered, disturbed, or annoyed”), and with an 11-point numerical scale. These verbal and numerical scales are proposed by ICBEN [5] and adopted in the International Organization for Standardization/technical specifications (ISO/TS) 15666 [6]. In addition, the annoyance ratings with the ICBEN modifiers and slightly different annoyance descriptors (“bothered”, “annoyed”, “noisy”, or “bothered or annoyed”) were included in eight datasets. Sato et al. [14] found no systematic difference in the relationship between exposure and annoyance with the ICBEN modifiers among four annoyance descriptors, using each of those prepared in JPN012CR2001 and JPN014CR2002. Therefore, we regarded that the annoyance with the ICBEN modifiers, regardless of annoyance descriptor, was equivalent to that measured on ICBEN five-point verbal scale. It should be noted that annoyance was not rated on the 11-point numerical scale in the above-mentioned eight datasets. Thus, annoyance measured on the five-point verbal scale is applied in the present analyses.

Before 2000, the following descriptors of annoyance were used: annoyed, unbearable, or dissatisfied. Regarding the number of scale points, except for the parts of JPN003RT1994, a four- or five-point scale was used. For example, the datasets of JPN003RT1994, JPN005RT1996, and JPN007RT1997, measured annoyance on the following four-point scale: “not at all”, “somewhat”, “significantly”, and “extremely” annoyed. On the other hand, annoyance for the datasets of JPN004HR1995, JPN006CR1997, and JPN009RT1998 was rated on the following five-point scale: “troublesome”, “bearable”, “slightly bearable”, “slightly unbearable”, and “unbearable”. JPN010RT1999 evaluated road traffic noise on the following unique category: “satisfied”, “a little satisfied”, “difficult to determine”, “a little dissatisfied”, and “dissatisfied”.

### 2.3. Noise Exposure

Table 2 shows noise exposure provided in each dataset. In this paper, the equivalent continuous A-weighted energy-equivalent sound pressure levels during daytime (from 7:00 to 19:00), evening (from 19:00 to 22:00) and nighttime (from 22:00 to 07:00) are defined as *L*_day_, *L*_evening_, and *L*_night_, respectively. *L*_Aeq,15h_ is the equivalent continuous A-weighted energy-equivalent sound pressure levels from 7:00 to 22:00. Noise exposure in *L*_den_ to each respondent, rounded to the nearest integer, was calculated based on the above three energy-equivalent sound pressure levels.

*L*_den_ was directly available for twelve of the thirty-four datasets. It should be noted that JPN010RT1999 includes micro-data with *L*_den_ in 1999 and those with *L*_Aeq,24_ in 2000. Therefore, we transformed *L*_Aeq,24_ values in 2000 using the following formula: *L*_den_ = *L*_Aeq,24h_ + 5.6 dB, given by the averaged difference between *L*_den_ and *L*_Aeq,24h_ values in 1999. In case of nine datasets (RT, six datasets; CR, three datasets), both metrics of *L*_Aeq,24h_ and *L*_dn_ are expressed by functions of *L*_Aeq,15h_ and *L*_night_. The *L*_Aeq,15h_ and *L*_night_ values can be obtained by solving the following equations (Equations (1) and (2)):(1)$$L_{\mathrm{Aeq},24h}=10\log_{10}\left\{\frac{15\times 10^{\frac{L_{\mathrm{Aeq},15h}}{10}}+9\times 10^{\frac{L_{\mathrm{night}}}{10}}}{24}\right\}$$
(2)$$L_{\mathrm{dn}}=10\log_{10}\left\{\frac{15\times 10^{\frac{L_{\mathrm{Aeq},15h}}{10}}+9\times 10^{\frac{L_{\mathrm{night}}+10}{10}}}{24}\right\}$$

Moreover, taking the number of trains in operation and road traffic volume in Japanese urban cities into account, we regarded that *L*_day_ and *L*_evening_ values were equal. As for road traffic noise, the difference between *L*_day_ and *L*_evening_ was confirmed using the two datasets (JPN007RT1997 and JPN010RT1999) for which both *L*_day_ and *L*_evening_ were available. As a result, the difference between both metrics was found to be 1.2 dB on average. However, a difference of about 1 dB between *L*_day_ and *L*_evening_ brought about a difference of only 0.2 dB in the *L*_den_ calculation. Therefore, we judged that *L*_day_ equals *L*_evening_ in six RT datasets. Likewise for railway noise, the difference between *L*_day_ and *L*_evening_ was confirmed using the three datasets (KMM102CR2009, KMM103CR2011 and KMM108CR2016) for which both *L*_day_ and *L*_evening_ were available. The difference between both metrics was found to be 0.8 dB on average. Therefore, we judged that *L*_day_ equals *L*_evening_ in three CR datasets. Based on each of the estimated *L*_day_, *L*_evening_, and *L*_night_ values, we calculated *L*_den_.

Considering the other road traffic datasets, noise exposures were related to *L*_Aeq,15h_ for JPN011RT2000 and JPN021RT2004. Based on the measurements of noise from the surveyed road disclosed by local governments, these values were converted to *L*_den._

Only *L*_Aeq,15h_ was originally provided in JPN017CR2003. Assuming that *L*_day_ and *L*_evening_ values were equal as previously mentioned, we set the values of *L*_day_ and *L*_evening_. Then, based on the number of trains operating from 7:00 to 22:00 (aggregated daytime and evening) and that from 22:00 to 7:00 (nighttime), we estimated *L*_night_ and *L*_den_. Other conventional railway datasets, JPN006CR1997 and JPN021CR2004, provided only *L*_Aeq,24h_. The correction of noise metrics was done not by referring to the timetable but from the averaged difference between *L*_den_ and *L*_Aeq,24h_ values given in JPN017CR2003. We corrected *L*_Aeq,24h_ values using the following formula: *L*_den_ = *L*_Aeq,24h_ + 4.7 dB.

Noise exposures related to *L*_Aeq,24h_ and *L*_Aeq,15h_ are available for five datasets for Shinkansen railway noise. Referring to the timetable on the target line, we examined the number of trains operating during each time category. In proportion to the number of trains by each time category, we estimated the energy-equivalent sound pressure levels (*L*_day_, *L*_evening_, and *L*_night_) and calculated *L*_den_.

Regarding the datasets of aircraft noise, JPN019CA2003 and JPN020MA2003, the monitoring data on number of flights measured around some surveyed airports/airbases and disclosed by local government are available. *L*_den_ was estimated based on the number of measurements for each time category. As for the surveyed airports of JPN019CA2003, where monitoring data were not disclosed, we decided that *L*_night_ was 0 dB, because of flight limitations during nighttime at the airports. We transformed the corresponding *L*_Aeq,24h_ values calculated from the original *L*_Aeq,15h_ and *L*_night_ into *L*_den_ using the following formula given in the previous report [31]: *L*_den_ = *L*_Aeq,24h_ + 1.5 dB. Likewise, some monitoring data around the surveyed airbases of JPN020MA2003 were not disclosed. Because of the irregular flights of military aircraft, unlike the regular flights of civil aircraft, it is difficult to estimate *L*_evening_ and *L*_night_ accurately. Thus, we excluded the corresponding data from the analyses. Finally, JPN024CA1996 provided *WECPNL* because the noise index was used in the former EQS for aircraft noise. In Japan, *WECPNL* was approximately calculated from *L*_A,Smax_, maximum A-weighted sound pressure level, instead of perceived noise level (*PNL*). Therefore, based on the relationship *PNL* ≈ *L*_A,Smax_ + 13, we transformed *WECPNL* values with the flowing formula: *L*_den_ = *WECPNL* − 13 dB [32].

## 3. Results

### 3.1. Demographic Factors and Noise Exposure

Table 3 shows the frequency distributions of demographic factors and housing types by the noise source. Values in brackets show the relative frequency. Except for the military aircraft noise, there were proportionally more female respondents than male. As for military aircraft noise (MA), males made up the majority of respondents. Respondents aged 60 years or older accounted for 36–54% for each noise source. On the other hand, respondents aged less than 40 years accounted for only 15–21%. The lower ratio of respondents aged 40 or less can be related to the fact that those who are living in detached houses accounted for over 70% of every dataset. About 90% of the detached houses had wooden frame structures.

To distinguish *L*_den_ in 5-dB steps from that in 1-dB steps, the former stands for *DENL* in this paper. Taking the estimation accuracy of low-level and high-level exposures and the current status of noise environment into account, we excluded data with *DENL* ≤ 30 dB or *DENL* ≥ 80 dB. Table 4 displays frequency distributions of *DENL* from 35 to 75 dB *L*_den_ for each transportation noise. For example, 50 dB *DENL* ranges from 48 dB to 52 dB *L*_den_. Values given in parentheses represent the percentages of the corresponding items. The numbers of *DENL* ≤ 30 dB and ≥80 dB were as follows: RT, 0 and 118; CR, 227 and 101; HR, 210 and 7; CA, 0 and 0; MA, 330 and 9.

Shinkansen railway noise (HR) had more respondents than other noises from 35 to 50 dB *DENL*. This tendency was particularly noticeable among datasets after 2010. Numbers of trains in operation on new projected Shinkansen lines where the surveys were carried out after 2010 were at a maximum of 140 trains per day. In addition, effective countermeasures against noise from new projected Shinkansen railways resulted in lower exposures than prior Shinkansen railways before 2010. Likewise, both civil and military aircraft noises (CA and MA), showed the tendency of lower noise exposure than road traffic and conventional railway noises. On the other hand, the road traffic noise (RT) showed the highest noise exposure. The percentage of *DENL* from 65 to 75 dB exceeded the majority (55%).

### 3.2. Overview of Exposure–Annoyance Relationships

In this section, we give an overview of the exposure–annoyance relationship for each of the 34 datasets. Schultz [33] used the term %HA to define the ratio of people who responded to either of the top two categories of a seven-point scale (cut-off point at 71%) or the top three categories of an 11-point scale (cut-off point at 73%). Miedema and Vos [34] defined the upper 28% of annoyance scales (cut-off point at 72%) as %HA, assuming that the interval scale between two consecutive ratings was equidistant regardless of different modifiers of annoyance and different scale points. In addition, most studies analyzed in the Guidelines used a cut-off point at 75% for a four-point scale or 60% and 72% for a five-point scale to define %HA. Based on the methods described in the abovementioned papers and guidelines, this study set the following cut-off point: 71% for the seven-point scale; 72% for the five-point and six-point scales; and 75% for the four-point scale. Therefore, we converted respondent’s rating with the score shown in Table 5 for each point scale. For example, in the five-point scale, similar to the above assumption that the annoyance scale is equidistant, we regarded all the respondents in the top category and 40% of respondents in the top second category as people highly annoyed by the main noise source. Likewise, for example, in the four-point scale, which is not rated on the ICBEN verbal scale, we regarded all the respondents in the top category as highly annoyed people. Thus, the averaged value of the converted scores in each of the exposure levels provide %HA at a cut-off point of around 72%.

Table 6 shows the association between *DENL* and %HA for each dataset. It should be noted that Table 6 is sorted in chronological order for each transportation noise. In the case of sample sizes of less than 25 (hyphen in the table), we did not calculate %HA. Table 6 shows that remarkable variance between datasets is observed even for the same transportation noise.

For road traffic noise, %HA values for JPN010RT1999 differs considerably. The %HA values are 50% and 74% at 55 dB and 65 dB, respectively. These percentages are 30% or higher than other datasets. In addition, 4 datasets from JPN003RT1994 through JPN009RT1998 have lower %HA than those after JPN011RT2000. The average value of %HA of the former datasets at 60 dB is approximately 9%, whereas that of the latter datasets is approximately 21%.

For conventional railway noise, %HA values for JPN002CR1994 and JPN006CR1997 whose annoyance rating is not measured on the ICBEN five-point verbal scale, are lower than those for datasets obtained after 2000. In particular, JPN006CR1997 shows no highly annoyed people at any valid exposure levels. Low percentages are likely attributed to the “unbearable” descriptor in this dataset.

Datasets for Shinkansen railway noise have more respondents than other noise sources at *L*_den_ value of 50 dB or less. At 50 dB, %HA values of Shinkansen railway noise range between 5 and 39. JPN004HR1995 employing “unbearable” as an annoyance descriptor shows slightly lower %HA. This tendency is observed in the datasets JPN006CR1997 and JPN009RT1998, which use the same “unbearable” descriptor. In addition, %HA values in datasets after 2010 (studies on new projected Shinkansen line), are about 10%HA lower compared to those before 2010.

%HA for civil aircraft noise at 55 dB varied from 2 to 44. JPN024CA1996 originating from the survey aimed to clarify the change in community response resulting from decreased exposure to aircraft noise. Therefore, excess response was likely to bring about lower %HA. Likewise, JPN020MA2003 shows much higher annoyance than JPN106MA2014. Both datasets used the ICBEN five-point verbal scale.

From the above findings, the effect of different scale from the ICBEN five-point verbal scale on %HA is likely to differ by the annoyance descriptor and the number of scale points. In addition, a surveyed period over 20 years can cause the change in community response to noise. Thus, we used micro-data with the ICBEN five-point verbal scale and with *DENL* range of 35 to 75 dB for next analysis.

### 3.3. Effect of Demographic Factors and Housing Type on %HA

Contributing to the establishment of the representative exposure–annoyance relationship, we examined the effects of demographic factors and housing type on noise annoyance. We applied logistic regression analysis to the datasets for each transportation mode. Highly annoyed response due to noise at a cut-off point of 72% was set as the dependent variable, while exposure level (*DENL*), gender (male and female), age (<40, 40–59, and ≥60), and housing structure (detached and apartment houses) were included as independent variables. In particular, *DENL* was not used as a categorical variable but as a continuous variable.

Based on the method created by Miedema and Vos [34], we defined all the responses in the top category (Category 5) of the ICBEN five-point verbal scale and 40% of responses in the second-to-top category (Category 4) as highly annoyed responses by the main specific noise source. Therefore, application of the logistic regression analysis requires converting the responses in the Category 4 into highly annoyed responses or not. In this section, we randomly divided the responses in the Category into two groups: 40% (HA) and 60% (not HA), following the Schreckenberg’s method [35]. Although this method is equivalent to Miedema and Vos, it is useful when applying logistic regression analysis to micro-data directly in the present study. Table 7 presents the results of multiple logistic regression analysis. The area under the curve (AUC) in Table 7 is larger than 0.7, except for Shinkansen railway noise. OR denotes the odds ratio based on the following reference category: male (gender), 40–59 (age), and detached house (housing type). 95% LCI and UCI stand for lower and upper limits of a 95% confidence interval, respectively. According to a rough calculation, the OR per 5 dB change in *DENL* equals around 1.8, regardless of transportation noise.

Every transportation noise shows the odds ratio of “female” is less than 1. In addition, the upper limit of a 95% confidence interval is lower than 1 for Shinkansen railway, civil aircraft, and military aircraft noises. Odds ratio of age category “≥60” is significant at a 10% level except for road traffic noise; however, age category “<40” shows no systematic difference. Odds ratio of “apartment house” is lower than 1 and significant at a 5% level except for military aircraft noise. Therefore, respondents living in detached houses are probably more annoyed than those living in apartments.

### 3.4. Establishment of Representative Exposure–Annoyance Relationship

The difference between detached and apartment houses in %HA was significant at a 5% level except for military aircraft noise. In addition to this, detached houses account for over 70% of all the respondents. Therefore, we establish Japanese representative exposure–annoyance relationship by transportation noise, derived from micro-data with the ICBEN verbal scale, *DENL* of 35–75 dB, and detached houses. Table 8 shows the observed %HA and sample size for each of *DENLs* and transportation noises. In the table, hyphen cells where the corresponding sample size is less than 50 and 100 responses for civil aircraft noise and other four individual noises, respectively, were excluded to establish the relationship.

Applying a quadratic regression to the relationship between *DENL* and %HA shown in Table 8, we plotted the modeled exposure–annoyance curve (solid line) and the 95% confidence interval curve (dot line) by transportation noise, together with observed %HA (gray circle) which are derived from aggregated datasets and data points derived from original micro-data of each dataset in Figure 1. No data point was plotted for ranges containing fewer than 25 responses. The equations for estimated %HA by *DENL* (5-dB steps) of each transportation noise are provided in Equations (3) to (7):
Estimated %HA of RT = 94.014 − 4.304 × *DENL* + 0.051 × *DENL*^2^ (*R*^2^ = 0.980) (3)
Estimated %HA of CR = 40.920 − 2.476 × *DENL* + 0.038 × *DENL*^2^ (*R*^2^ = 0.996) (4)
Estimated %HA of HR = 35.396 − 2.482 × *DENL* + 0.043 × *DENL*^2^ (*R*^2^ = 0.966) (5)
Estimated %HA of CA = 135.705 − 6.041 × *DENL* + 0.076 × *DENL*^2^ (*R*^2^ = 0.988) (6)
Estimated %HA of MA = −68.080 + 1.838 × *DENL* + 0.006 × *DENL*^2^ (*R*^2^ = 0.994) (7)

In addition, Table 9 lists the estimated %HA and 95% confidence interval (lower and upper) of the modeled quadratic regression.

The estimated %HA for military aircraft noise is highest, followed by civil aircraft noise. At each range from 45 to 60 dB, the estimated lower limit of a 95% confidence interval of military aircraft noise is higher than the estimated upper limit of a 95% confidence interval of civil aircraft noise. An estimated 95% confidence interval of Shinkansen railway noise is broader than those of other transportation noises. This is related to a noticeable difference in exposure–annoyance relationship between projected Shinkansen lines (Hokuriku and Kyushu Shinkansen lines) and prior Shinkansen lines, such as the Tokaido and Sanyo Shinkansen lines. In contrast, the estimated 95% confidence interval of conventional railway noise is the lowest. In addition, at each range from 55 to 70 dB, the estimated lower limit of a 95% confidence interval of conventional railway noise is higher than the estimated upper limit of a 95% confidence interval of road traffic noise. This means %HA due to conventional railway noise is significantly higher at a 5% level than that due to road traffic noise.

### 3.5. Comparison of Estimated Exposure–Annoyance Curves

Figure 2 compares estimated exposure–annoyance curves derived from the Japanese dataset with those derived from the WHO dataset (Guidelines). Considering the numbers of datasets and sample sizes related to aircraft noises, we address road traffic and railway noises.

The estimated %HA values of Japanese and WHO road traffic noise at around 54 dB are almost equal and the %HA for Japanese dataset gets higher with increasing *L*_den_ than the WHO dataset. For example, estimated %HA values at 60 dB are 19.4 and 15.1 %HA for Japanese and WHO datasets, respectively; while those at 70 dB are 42.6 and 28.4 %HA. In the case of railway noise, estimated %HA for each of the Japanese datasets is higher than that for the WHO dataset. On average, the %HA for Japanese conventional railway noise is 11.0 %HA higher than that for WHO railway noise. Likewise, the percentage of high annoyance for Shinkansen railway noise is, on average, 12.7 %HA higher than the WHO railway noise within levels between 40 to 60 dB.

## 4. Discussion

Although the number of aircraft noise datasets available for analysis was few, Figure 1f and Table 9 show that the estimated prevalence of highly annoyed people due to military aircraft noise was the highest among transportation noises, followed by civil aircraft noise. This was consistent with the relationship given by Miedema and Voss [34] and the Guidelines [1], that the percentage of highly annoyed people due to aircraft noise was higher compared to ground transportation noises. In particular, military aircraft noise led to extreme proportions. Morinaga et. al. showed higher annoyance for military aircraft noise than that for civil aircraft noise when the noise levels were equal [28]. It was also found that fluctuation of daily flight numbers affected long-term evaluation around military airfields. Yamada et al. [36] reported that residents around military airfields have been exposed to much larger noise levels than the annual average value occasionally because of a large irregular change in the total of aircraft movements from day to day. As another former study reported [37], the long-term impression of environmental noise was judged on the basis of memory of prominent sounds. It contributes greatly to the determination of a long-term impression; the memory of occasional high-level noise events has the possibility to increase annoyance. Although the percentage of highly annoyed people due to civil aircraft noise is the second highest, a broad confidence interval brings about no statistically significant difference from Shinkansen railway noise.

The percentage of highly annoyed people due to Shinkansen railway noise was higher than that due to conventional railway noise or road traffic noise. For example, Yokoshima and Tamura [9] clarified that annoyance due to noise from Shinkansen railway was higher than that from conventional railway. Yano et al. [17] also showed that the prevalence of noise annoyance and vibration annoyance in areas along Shinkansen railway were high. This paper focused on the following factors: vertical ground-borne vibrations (building vibrations) and attitudes towards the noise source.

Yokoshima et al. confirmed that vertical ground-borne vibration from Shinkansen railway was higher than conventional railway or road traffic vibration at the same sound pressure level [38,39]. Likewise, it can be thought that residents living in detached houses along Shinkansen railways are exposed to larger building vibrations than those along conventional railways or trunk roads. In terms of the effect of vibrations on noise annoyance under such situations, Yokoshima and Tamura clarified the combined effect of vertical ground-borne vibration on noise annoyance only in detached houses along Shinkansen railways [15]. Moreover, Yokoshima et al. confirmed the significantly increasing effect of vertical ground-borne vibration on noise annoyance, aggregating Japanese micro-data including annoyance and exposure associated with each of noise and vibration [40]. These findings reveal the combined effect increases the prevalence of highly annoyed people due to Shinkansen railway noise compared to other ground transportations. Tamura [41], Sato [42], and Pedersen et al. [43], reported that the attitudes towards the noise source were associated with noise annoyance. In particular, Tamura [41] conducted socio-acoustic surveys and in-depth interviews in areas exposed to Shinkansen railway noise or conventional railway noise. Noise from Shinkansen railway was regarded more negatively than conventional railway noise. He also indicated that residents living in the areas along Shinkansen railway were generally concerned with noise issues and recognized no need for the Shinkansen railway noise. Under such circumstances, it can be assumed that an effect of building vibrations on noise annoyance associated with Shinkansen railway is likely to be larger for residents living in the vicinity of the railway.

The percentage of highly annoyed people due to conventional railway noise was higher than that due to road traffic noise reported in the previous Japanese studies [3,4]. Thus, socio-acoustic surveys in Japan have revealed no railway bonus. Morihara et al. [44] showed that the distance from transportation to the dwelling was one of the factors causing the difference in exposure–response relationships for railway and road traffic noises between Euro-American countries and Japan. For example, building vibrations that we have already focused on are expressed by a function of the distance. Therefore, there is a possibility that the perception of building vibrations increase annoyance due to noise from the conventional railway because the conventional railway is more likely to generate larger ground-borne vibrations than road traffic [38]. Gidlöf-Gunnarsson [45] also showed that the proportion annoyed by railway noise in areas where the railway traffic causes strong ground-borne vibrations and in areas with a very large number of trains was higher than in areas without vibration. The high frequency of conventional trains in Japan can cause a higher proportion of highly annoyed people due to noise, compared with road traffic noise. Likewise, Ögren et al. [46] indicated that the vibration velocity influenced annoyance from railway noise. Peris et al. [47] also concluded that neglecting vibrations results in an underestimation of people highly annoyed by noise. People noticing railway vibration are more likely to be highly annoyed by railway noise than people who do not notice vibration. Moreover, the progress of countermeasures differs between road traffic and conventional railway noises. Various measures, such as reduction of noise produced by the operation of motor vehicles, reduction of drainage pavement noises, and installation of sound insulation walls against road traffic noise have been taken for many years. In contrast, no regulations or standards concerning noise from existing conventional railways have been established in Japan. It could not be denied that higher annoyance due to railway noise is attributed to the difference in the circumstances surrounding individual transportations.

The exposure–annoyance relationship of ground transportation noise in the Japanese dataset was considerably higher than that in the WHO dataset. The Japanese exposure–annoyance relationships were established based on the annoyance of residents living in detached houses. The sound insulation performance of detached houses made of wooden structure is lower than that of apartment houses made of reinforced concrete. In addition, the difference in building foundations and structure between detached and apartment houses also brings about the difference in vibration-proofing performance. Therefore, even though noise exposures at the facade of dwellings are the same level, residents in detached houses are exposed to larger noise than those in apartment houses. The difference in exposure to indoor noise is a factor constituting higher annoyance in the Japanese dataset. Similarly, higher annoyance for railway noise than that for road traffic noise, was obtained from a recent re-analysis in Korea [48]. However, the most prevalent housing type at the railway and road noise survey sites are multi-story apartments, which is the most common type of residence in Korea. This finding suggests that factors except for building vibrations also relate to the difference in annoyance between railway and road traffic noises. Noise annoyance is a multifaceted psychological construct that is affected by the above vibration and attitudes and the following factors: the distance between transportation and house, visibility, demographic factors, lifestyle, properties of the survey area, nationality, and so forth [49].

In Japan, whether the guideline exposure levels can be applied to standard values for the EQSs for the noises or not is currently under discussion. The EQSs for environmental noises are indispensable for planning noise policy and creating effective countermeasures. The EQSs shall be established or revised whenever necessary, given that new scientific knowledge of environmental noise effects on humans is presented. Since annoyance in Japanese datasets is higher than that in WHO datasets, the Japanese unique exposure–annoyance relationships for each noise source obtained in this paper contributes to the formation of an effective noise policy.

The applications of the findings in this study are limited because the results are based on the annoyance due to transportation noise only in Japanese detached houses. Therefore, annoyance in apartment houses differs from the results obtained in this paper. Even in case of detached houses that are made with a steel frame or reinforced concrete, noise annoyance can be lower than that in wooden detached houses.

Finally, following the publication of the WHO, Van Kamp [50] collected new exposure–response relationships, published between January 2014 and December 2019, and updated the exposure–response relationships for annoyance. However, given that each country has a different cultural background, establishing representative exposure–response relationships in each country is an important issue in discussing noise policy. As for Asian countries except for Japan, Nguyen et al. [51] in Vietnam and Hong et al. [48] in South Korea, reported representative exposure–annoyance relationships related to transportation noise, using social–acoustic survey data accumulated in each country. In China as well, Guoqing et al. [52], Zhang and Ma [53], and Xie et al. [54] reported community response to aircraft noise, high-speed railway noises, and road traffic noise, respectively. In the future it is expected that more useful findings, including the effects of non-acoustic factors (e.g., economic factors and topology [54]), can be obtained if the accumulation of data is further advanced and representative exposure–annoyance relationships in each country can be compared.

## 5. Conclusions

This paper aims to establish the representative relationship between noise exposure and the prevalence of highly annoyed people due to transportation noise in Japan. We accumulated thirty-four datasets, which were provided by SASDA and derived from the other surveys conducted in Japan. All the datasets include the following micro-data: demographic factors, exposure and annoyance data associated with specific transportation noise. Using micro-data with the ICBEN five-point verbal scale, 5-dB steps from 35 to 75 of *L*_den_, and detached houses, we performed a secondary analysis and established modeled exposure–annoyance relationships by transportation noise.

Among the five transportation noises, military aircraft noise has the highest proportion of high annoyance, followed by civil aircraft noise. Shinkansen railway noise showed a higher proportion compared to conventional railway noise and road traffic noise. To validate the representativeness of the modeled exposure–annoyance curves, we discussed factors affecting the difference in annoyance. In addition, the comparison of modeled exposure–annoyance relationships between the Japanese and WHO datasets revealed the Japanese annoyance is higher than the WHO annoyance. Furthermore, we discussed non-acoustic factors causing the difference in exposure–annoyance relationship between Japanese and other counties.

Contributing these results to the review of the Japanese present standards and guidelines, we need to approach the Japanese government. To utilize these results more effectively in noise policy, it is also necessary to introduce the percentage of highly annoyed people due to noise as an index for a solution when dealing with noise problems. Updating the modeled exposure–annoyance relationship requires annoyance due to every transportation noise in apartment houses. In addition, as for the issues to be solved from now on, we need to accumulate datasets associated with civil and military aircraft noises. In particular, as the modeled exposure–annoyance relationship related to civil aircraft noise varied widely, it is important to enhance the accuracy of the modeled relationship. Likewise, for Shinkansen railway noise, the difference in annoyance due to noise between new projected and original Shinkansen lines should be analyzed. Moreover, discussion on whether Shinkansen railway noise annoyance is increasing or decreasing over time is expected to deepen.

## Figures and Tables

**Figure 1 ijerph-18-10935-f001:**
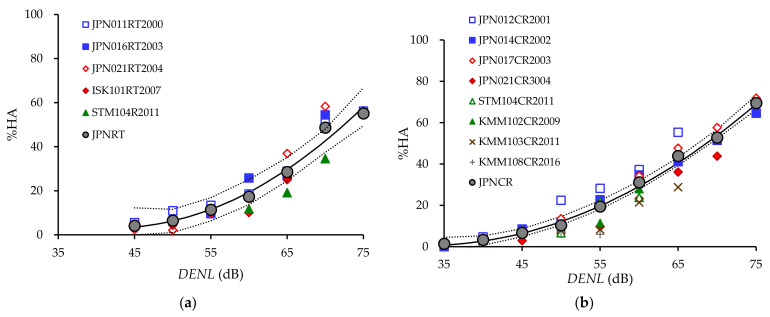
Estimated exposure–annoyance relationships (line) derived from aggregated data and data-points derived from micro-data: (**a**) road traffic noise; (**b**) conventional railway noise; (**c**) Shinkansen railway noise; (**d**) civil aircraft noise; (**e**) military aircraft; (**f**) comparison among transportation noises. Figure 1 (**a**–**e**) were plotted in 5-dB units of the noise level; Figure 1 (**f**) was drawn by interpolating the noise level in 1-dB units.

**Figure 2 ijerph-18-10935-f002:**
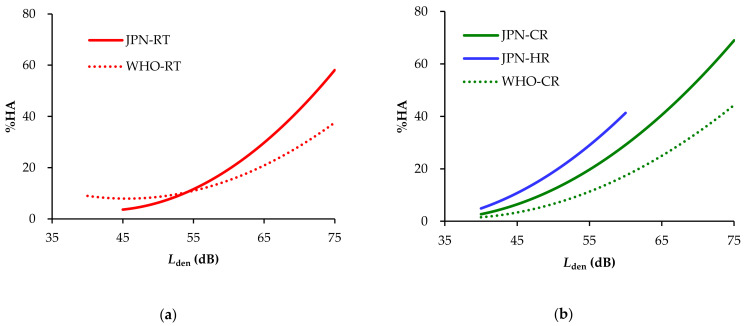
Comparison between estimated exposure–annoyance curves derived from Japanese and WHO datasets: (**a**) Japanese road traffic noise (JPN-RT) vs. WHO road traffic noise; (**b**) Japanese conventional railway noise (JPN-CR) and Shinkansen railway noise of (JPN-HR) vs. WHO railway noise WHO-Rail). Both figures were drawn by interpolating the noise level in 1-dB units.

**Table 1 ijerph-18-10935-t001:** Outline of datasets.

No	Dataset	Source	Survey Year	Survey Site	Sample Size
1	JPN002CR1994 [7]	CR	1994–1995	Kumamoto Pref.	1828
2	JPN003RT1994 [8]	RT	1994–1995	Kumamoto & Fukuoka Prefs.	387
3	JPN004HR1995 [9]	HR	1995–1996	Kanagawa Pref.	864
4	JPN005RT1996 [10]	RT	1996	Kumamoto Pref.	816
5	JPN006CR1997 [9]	CR	1997	Kanagawa Pref.	308
6	JPN007RT1997 [11]	RT	1997–1998	Sapporo City	780
7	JPN009RT1998 [9]	RT	1998	Kanagawa Pref.	353
8	JPN010RT1999 [12]	RT	1999–2000	Kanagawa Pref.	652
9	JPN011RT2000 [13]	RT	2000–2006	Saitama, Chiba, Tokyo, Kanagawa, Nagano, Osaka, and Fukuoka Prefs.	1601
10	JPN012CR2001 [14]	CR	2001	Hokkaido Pref.	1442
11	JPN013HR2001 [15]	HR	2001–2003	Kanagawa Pref.	1101
12	JPN014CR2002 [16]	CR	2002	Fukuoka Pref.	1579
13	JPN015HR2003 [17]	HR	2003	Fukuoka Pref.	715
14	JPN016RT2003 [18]	RT	2003–2004	Hokkaido Pref.	272
15	JPN017CR2003 [19]	CR	2003–2006	Chiba, Tokyo, Kanagawa, Aichi, Osaka, and Kumamoto Prefs.	1490
16	JPN018HR2003 [19]	HR	2003–2006	Tochigi, Saitama, Tokyo, Kanagawa, Nagano, Shizuoka, and Osaka Prefs.	1306
17	JPN019CA2003 [19]	CA	2003–2006	Miyagi, Tokyo, Osaka, and Fukuoka Prefs.	850
18	JPN020MA2003 [19]	MA	2003–2006	Ibaraki, Saitama, Tokyo, Kanagawa, and Fukuoka Prefs.	583
19	JPN021RT2004 (CB) [20]	RT	2004–2006	Kanagawa Pref.	1358
20	JPN021CR2004 (CB) [20]	CR	2004–2006	Kanagawa Pref.	1357
21	JPN022HR2005 [21]	HR	2005	Nagoya City	174
22	JPN023CA2006 [22]	CA	2006	Kumamoto Pref.	411
23	JPN024CA1996 [23]	CA	1996	Osaka Pref.	208
24	ISK101RT2007 [24]	RT	2007	Ishikawa Pref.	371
25	KMM102CR2009 [25]	CR	2009–2010	Kumamoto Pref.	601
26	KMM103CR2011 (CB) [25]	CR	2011–2012	Kumamoto Pref.	1028
27	KMM103HR2011 (CB) [25]	HR	2011–2012	Kumamoto Pref.	1031
28	STM104RT2011 [26]	RT	2011	Saitama City	189
29	STM104CR2011 [26]	CR	2011	Saitama City	162
31	NGN105HR2013 [27]	HR	2013	Nagano Pref.	293
30	JPN106MA2014 [28]	MA	2014	Ibaraki, Tochigi, Tokyo, Shizuoka, Ishikawa, Yamaguchi, and Kagoshima Prefs.	4264
32	HKR107HR2016 [29]	HR	2016	Ishikawa & Toyama Prefs.	919
33	KMM108CR2016 (CB) [30]	CR	2016–2017	Kumamoto Pref.	693
34	KMM108HR2016 (CB) [30]	HR	2016–2017	Kumamoto Pref.	691

**Table 2 ijerph-18-10935-t002:** List of the number of scale points and descriptor of annoyance due to specific noise source and noise exposure.

Dataset	Number of Scale Points and Descriptor of Annoyance	Exposure
JPN002CR1994	four-/five-/six-/seven-point verbal scales (annoyed)	*L*_Aeq,24h_/*L*_dn_
JPN003RT1994	four-point verbal scale (annoyed)	*L*_Aeq,24h_/*L*_dn_
JPN004HR1995	five-point verbal scale (endured)	*L* _Aeq,24h_
JPN005RT1996	four-point verbal scale (annoyed)	*L*_Aeq,24h_/*L*_dn_
JPN006CR1997	five-point verbal scale (endured)	*L* _Aeq,24h_
JPN007RT1997	four-point verbal scale (annoyed)	*L* _den_
JPN009RT1998	five-point verbal scale (endured)	*L*_Aeq,24h_/*L*_dn_
JPN010RT1999	five-point verbal scale (dissatisfied)	*L*_den_/*L*_Aeq,24h_
JPN011RT2000	ICBEN five-point verbal scale (bothered, disturbed, or annoyed)	*L* _Aeq,15h_
JPN012CR2001	four-point verbal scale (annoyed) four-point verbal scale (bothered, disturbed, or annoyed) ICBEN five-point verbal scale (bothered, disturbed, or annoyed)	*L*_Aeq,24h_/*L*_dn_
JPN013HR2001	ICBEN five-point verbal scale (bothered)	*L* _Aeq,24h_
JPN014CR2002	ICBEN five-point verbal scale (bothered, disturbed, or annoyed) ICBEN five-point verbal scale (bothered) ICBEN five-point verbal scale (annoyed) ICBEN five-point verbal scale (noisy)	*L* _den_
JPN015HR2003	ICBEN five-point verbal scale (bothered, disturbed, or annoyed)	*L* _Aeq,24h_
JPN016RT2003	ICBEN five-point verbal scale (bothered)	*L*_Aeq,24h_/*L*_dn_
JPN017CR2003	ICBEN five-point verbal scale (bothered, disturbed, or annoyed)	*L* _Aeq,15h_
JPN018HR2003	ICBEN five-point verbal scale (bothered, disturbed, or annoyed)	*L* _day,15h_
JPN019CA2003	ICBEN five-point verbal scale (bothered, disturbed, or annoyed)	*L* _Aeq,15h_
JPN020MA2003	ICBEN five-point verbal scale (bothered, disturbed, or annoyed)	*L* _Aeq,15h_
JPN021RT2004	ICBEN five-point verbal scale (bothered)	*L* _Aeq,15h_
JPN021CR2004	ICBEN five-point verbal scale (bothered)	*L* _Aeq,24h_
JPN022HR2005	ICBEN five-point verbal scale (bothered or annoyed)	*L* _Aeq,24h_
JPN023CA2006	ICBEN five-point verbal scale (bothered, disturbed, or annoyed)	*L* _den_
JPN024CA1996	five-point scale (annoyed)	*WECPNL*
ISK101RT2007	ICBEN five-point verbal scale (bothered, disturbed, or annoyed)	*L*_Aeq,24h_/*L*_dn_
KMM102CR2009	ICBEN five-point verbal scale (bothered, disturbed, or annoyed)	*L* _den_
KMM103CR2011	ICBEN five-point verbal scale (bothered, disturbed, or annoyed)	*L* _den_
KMM103HR2011	ICBEN five-point verbal scale (bothered, disturbed, or annoyed)	*L* _den_
STM104RT2011	ICBEN five-point verbal scale (bothered or annoyed)	*L*_Aeq,24h_/*L*_dn_
STM104CR2011	ICBEN five-point verbal scale (bothered or annoyed)	*L*_Aeq,24h_/*L*_dn_
NGN105HR2013	ICBEN five-point verbal scale (bothered, disturbed, or annoyed)	*L* _den_
JPN106MA2014	ICBEN five-point verbal scale (bothered, disturbed, or annoyed)	*L* _den_
HKR107HR2016	ICBEN five-point verbal scale (bothered, disturbed, or annoyed)	*L* _den_
KMM108CR2016	ICBEN five-point verbal scale (bothered, disturbed, or annoyed)	*L* _den_
KMM108HR2016	ICBEN five-point verbal scale (bothered, disturbed, or annoyed)	*L* _den_

**Table 3 ijerph-18-10935-t003:** Frequency of distributions of demographic factors and housing type for each noise. DH and AH in housing type indicate detached house and apartment house, respectively.

Item	Gender			Age				Housing Type		
Category	Male	Female	No Answer	<40	41–59	60≤	No Answer	DH	AH	No Answer
RT	2886 (43%)	3798 (56%)	95 (1%)	1391 (21%)	2807 (41%)	2466 (36%)	115 (2%)	4861 (72%)	1719 (25%)	199 (3%)
CR	4242 (40%)	6152 (59%)	94 (1%)	1968 (19%)	4136 (39%)	4308 (41%)	76 (1%)	8261 (79%)	1882 (18%)	345 (3%)
HR	2867 (40%)	4135 (58%)	92 (1%)	1114 (16%)	2700 (38%)	3206 (45%)	74 (1%)	5927 (84%)	1067 (15%)	100 (1%)
CA	647 (44%)	783 (53%)	39 (3%)	243 (17%)	544 (37%)	676 (46%)	6 (0%)	1065 (72%)	311 (21%)	93 (6%)
MA	2527 (52%)	2276 (47%)	44 (1%)	713 (15%)	1465 (30%)	2631 (54%)	38 (1%)	3969 (82%)	828 (17%)	50 (1%)

**Table 4 ijerph-18-10935-t004:** Frequency distributions of noise exposure for each noise source.

	*DENL* (dB)	35	40	45	50	55	60	65	70	75
Noise Source										
RT		0 (0%)	16 (0%)	178 (3%)	476 (7%)	834 (13%)	1456 (22%)	1693 (25%)	1327 (20%)	681 (10%)
CR		365 (4%)	611 (6%)	733 (7%)	1458 (14%)	1963 (19%)	2063 (20%)	1766 (17%)	890 (9%)	311 (3%)
HR		205 (3%)	459 (7%)	1692 (25%)	2445 (36%)	1387 (20%)	467 (7%)	171 (2%)	39 (1%)	12 (0%)
CA		0 (0%)	51 (3%)	351 (24%)	201 (14%)	406 (28%)	202 (14%)	243 (17%)	14 (1%)	1 (0%)
MA		66 (1%)	162 (4%)	780 (17%)	1232 (27%)	847 (19%)	927 (21%)	322 (7%)	97 (2%)	75 (2%)

**Table 5 ijerph-18-10935-t005:** Converted score for each point scale.

Number of Scale Points	Category						
	1	2	3	4	5	6	7
four-point scale	0	0	0	1	-	-	-
five-point scale	0	0	0	0.4	1	-	-
six-point scale	0	0	0	0	0.7	1	-
seven-point scale	0	0	0	0	0	1	1

**Table 6 ijerph-18-10935-t006:** %HA as a function of *DENL* (5-dB step in *L*_den_) for each dataset.

Dataset	*DENL* (dB)								
	35	40	45	50	55	60	65	70	75
JPN003RT1994	-	-	-	11	7	11	18	-	-
JPN005RT1996	-	-	-	-	3	8	19	28	38
JPN007RT1997	-	-	-	-	-	5	10	27	37
JPN009RT1998	-	-	-	-	6	11	17	35	29
JPN010RT1999	-	-	-	-	50	55	74	81	91
JPN011RT2000	-	-	6	11	13	20	36	48	54
JPN016RT2003	-	-	-	-	10	26	27	54	-
JPN021RT2004	-	-	2	4	11	15	31	40	43
ISK101RT2007	-	-	4	4	12	10	25	-	-
STM104RT2011	-	-	-	-	-	12	19	35	-
JPN002CR1994	-	1	3	5	12	18	27	33	-
JPN006CR1997	0	0	0	0	-	-	-	-	-
JPN012CR2001	2	3	9	17	19	31	45	47	-
JPN014CR2002	0	3	9	11	23	35	41	52	65
JPN017CR2003	-	-	7	14	20	34	48	58	72
JPN021CR3004	-	4	3	7	10	23	36	44	58
STM104CR2011	-	-	-	-	8	21	29	-	-
KMM102CR2009	2	-	-	7	22	24	-	-	-
KMM103CR2011	0	3	8	9	11	28	-	-	-
KMM108CR2016	2	3	7	7	6	23	-	-	-
JPN004HR1995	2	3	12	16	16	24	32	-	-
JPN013HR2001	-	11	16	20	29	38	42	56	-
JPN015HR2003	-	11	12	39	37	-	-	-	-
JPN018HR2003	-	-	13	21	29	45	56	-	-
JPN022HR2005	-	-	-	15	21	15		-	-
KMM103HR2011	-	-	4	7	6	5	6	-	-
NGN105HR2013	0	3	5	11	-	-	-	-	-
HKR107HR2016	-	-	9	22	11	-	-	-	-
KMM108HR2016	1	5	4	5	-	-	-	-	-
JPN019CA2003	-	-	17	16	12	48	59	-	-
JPN023CA2006	-	17	15	-	44	-	-	-	-
JPN024CA1996	-	-	-	-	2	-	28	-	-
JPN020MA2003	-	-	-	69	75	72	89	91	89
JPN106MA2014	5	15	23	39	47	56	74	56	-

**Table 7 ijerph-18-10935-t007:** Exposure–annoyance relationships for each dataset. Odds ratio (OR) means a change of 1 dB in the noise level.

	Item	Category	Estimate	S.E.	*p*	OR	95% LCI	95% UCI
RT	*DENL*		0.115	0.006	0.000	1.122	1.108	1.136
*n* = 3688	Gender	Female	−0.090	0.088	0.309	0.914	0.769	1.087
AUC = 0.731	Age	<40	−0.268	0.133	0.044	0.765	0.589	0.992
		≥60	0.135	0.097	0.163	1.145	0.947	1.384
	Housing type	Apartment house	−0.330	0.108	0.002	0.719	0.582	0.889
	Constant		−8.309	0.408	0.000	0.000		
CR	*DENL*		0.115	0.004	0.000	1.122	1.113	1.131
*n* = 7092	Gender	Female	−0.100	0.065	0.121	0.905	0.797	1.027
AUC = 0.768	Age	<40	0.029	0.091	0.753	1.029	0.861	1.230
		≥60	−0.192	0.071	0.007	0.825	0.718	0.948
	Housing type	Apartment house	−0.447	0.077	0.000	0.639	0.550	0.743
	Constant		−7.612	0.250	0.000	0.000		
HR	*DENL*		0.109	0.006	0.000	1.115	1.102	1.129
*n* = 6165	Gender	Female	−0.159	0.071	0.025	0.853	0.743	0.980
AUC = 0.677	Age	<40	−0.070	0.106	0.511	0.932	0.757	1.148
		≥60	−0.251	0.077	0.001	0.778	0.668	0.906
	Housing type	Apartment house	−0.837	0.111	0.000	0.433	0.348	0.539
	Constant		−6.760	0.335	0.000	0.001		
CA	*DENL*		0.123	0.012	0.000	1.131	1.106	1.157
*n* = 1261	Gender	Female	−0.476	0.155	0.002	0.621	0.459	0.841
AUC = 0.752	Age	<40	−0.088	0.241	0.714	0.916	0.571	1.468
		≥60	0.399	0.167	0.017	1.491	1.074	2.069
	Housing type	Apartment house	−1.315	0.208	0.000	0.268	0.179	0.403
	Constant		−7.314	0.627	0.000	0.001		
MA	*DENL*		0.107	0.005	0.000	1.113	1.103	1.123
*n* = 4508	Gender	Female	−0.130	0.067	0.052	0.878	0.771	1.001
AUC = 0.708	Age	< 40	−0.001	0.105	0.992	0.999	0.814	1.227
		≥ 60	0.129	0.075	0.084	1.138	0.983	1.318
	Housing type	Apartment house	−0.089	0.090	0.321	0.915	0.768	1.091
	Constant		−5.876	0.266	0.000	0.003		

**Table 8 ijerph-18-10935-t008:** Observed %HA and sample size of *DENL* by transportation noise. Values in each cell are as follows: top, %HA; bottom, sample size.

Source	*DENL* (dB)								
	35	40	45	50	55	60	65	70	75
RT	-	-	4.1 136	6.5 312	11.4 575	17.3 670	28.5 496	48.6 294	55.1 159
CR	1.4 243	3.3 371	6.7 454	10.3 795	19.4 886	31.1 852	43.9 736	52.8 451	69.5 207
HR	0.7 120	5.0 284	9.0 1275	22.3 1985	23.9 1089	42.1 240	-	-	-
CA	-	16.5 51	15.8 333	21.1 130	36.2 193	45.4 118	62.8 64-	-	-
MA	-	14.8 134	24.5 609	40.8 998	50.0 719	60.8 761	76.5 259	-	-

**Table 9 ijerph-18-10935-t009:** Estimated %HA and 95% confidence interval of quadratic regression for 5-dB steps of *L*_den_ by transportation noise.

*DENL*	RT		CR		HR		CA		MA	
	%HA	95%CI	%HA	95%CI	%HA	95%CI	%HA	95%CI	%HA	95%CI
35			0.8	0.0–4.5	1.2	0.0–11.5				
40			2.7	0.7–5.2	4.9	0.0–11.1	15.7	7.8–22.9	15.0	8.1–21.0
45	3.6	0.0–12.3	6.5	4.8–8.7	10.8	3.2–17.5	17.8	12.7–22.0	26.8	22.2–30.1
50	6.3	1.1–11.6	12.1	10.4–14.5	18.8	11.2–25.4	23.7	18.0–28.2	38.8	33.7–42.4
55	11.6	6.3–16.9	19.7	18.0–22.3	29.0	21.9–34.8	33.4	27.6–37.8	51.2	45.9–54.6
60	19.4	13.7–25.1	29.2	27.6–31.7	41.3	29.9–51.1	46.8	41.4–50.7	63.8	58.8–66.7
65	29.7	24.5–35.0	40.5	39.2–43.1			64.1	55.7–70.8	76.7	69.0–82.0
70	42.6	37.4–47.9	53.8	52.3–56.8						
75	58.1	49.5–66.7	69.0	66.3–73.3						

## Data Availability

This study re-analyzes Socio-Acoustic Survey Data Archive (SASDA) at http://www.ince-j.or.jp/old/04/04_page/04_doc/bunkakai/shachodata/?page_id=972 (accessed on: 16 October 2021) and other datasets. Regarding the other datasets, those are not stored in SASDA, we obtained consent for the secondary analysis from the project manager of each survey.

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
