# Peer review of "Representative Exposure–Annoyance Relationships Due to Transportation Noises in Japan"

_ijerph, 2021, doi:10.3390/ijerph182010935_

Round 1

Reviewer 1 Report

The article is very technical and well structured. It can be accepted for publication. 

A review of written language and formatting can improve the text.

I point out, as examples, for the authors' evaluation:

Line 67: "were reported" instead of "was reported";

Line 229: "has more" instead of "have more";

Table 3: The first and the second-row mix bold and no bold;

Line 304: The reference should be [34] instead of [33];

Line 344: "are provided" instead of "is provided"; 

Line 416: The sentence "In general..." can be rewritten for better understanding;

Line 469: "relate" instead of "relates";

Line 501: "revealed" instead of "reveled".

Author Response

Thank you for your suggestions, please see the attachment to check the response.

Reviewer 2 Report

The authors present a study on the relationship between noise exposure level for each transportation noise and the percentage of highly annoyed people. They performed a secondary analysis by using more than 30000 micro-data based on 34 datasets in Japan. They found that the estimated prevalence of highly annoyed people due to military aircraft noise was the highest among transportation noises, followed by civil aircraft noise. The structure of this paper is fine, and the analysis of research data is well organized. However, the creativity of this research should be clarified, I recommend major revisions as follows:

  • What is the creativity of this research? In Line 68-75, “... the technical committee preformed a secondary analysis. It It showed the percentage of highly annoyed people (%HA) as a function of Ldn for each transportation noise...” “In this paper, adding the other datasets derived from recent studies conducted after 2007,....” What is the difference between this study and previous study? Is the data size the only difference? Why it is necessary to do this research?
  • Please update the references. There are only 5 references (3 in 2018, 1 in 2019, 1 ISO standard in 2021) in recent 3 years. It is important to decide whether this research is necessary or not.
  • Also, a large number of references are self-cited. It’s recommended to make more comparisons with other countries particularly in Section4, and include more relevant previous studies, such as
    ‘Noise exposure of the residential areas close to urban expressways in a high-rise mountainous city. Environment and Planning B, 2021‘
    ’Effects of socio-economic indicators on perceptions of urban acoustic environments in Chinese megacities. Indoor and Built Environment ,2020’
  • In Table 1, the authors give a number for each dataset, why not use this number instead of “JPN002CR1994”?
  • Several datasets are derived from a survey conducted in areas exposed to combined noise sources, I would recommend to show the combined noise sources in Table 1 as well.
  • Section 2.2: in some datasets, the annoyance was not measured on ICBEN 5-point scale. How did the authors deal with theses data? How to compare these data with ICBEN descriptors?
  • Line 177, Line 200: why Lday and Levening can be regarded as equal? Please show evidence.
  • What is meaning of “35, 40...” in the first row of Table 4? Does “35dB” DENL means 35dB<DENL<=40dB”?
  • The authors analyze the prevalence of highly annoyed people due to different noise sources: military aircraft is the highest, followed by civil aircraft... Did authors analyze the extent of the HA change with the increase of exposure level? (Figure 1f)

Author Response

(The authors gave the same response as above.)

Reviewer 3 Report

I have only a few remarks: 

line 304: wrong reference
line 416: check grammar - respondents are exposed to vibrations (and not the other way around)

Figur 2. I would recommend to use the same color for the WHO reference in both panels

Author Response

(The authors gave the same response as above.)
